# Snowfall in Northern Finland derives mostly from ice clouds

Claudia Mignani[1,a], Lukas Zimmermann[1], Rigel Kivi[2], Alexis Berne[3], and Franz Conen[1]

[1]Department of Environmental Sciences, University of Basel, 4056 Basel, Switzerland
[2]Space and Earth Observation Centre, Finnish Meteorological Institute, 99600 Sodankylä, Finland
[3]Environmental Remote Sensing Laboratory, Swiss Federal Institute of Technology in Lausanne, 1015 Lausanne, Switzerland
[a]now at: Department of Atmospheric Science, Colorado State University, Fort Collins, CO 80521, USA

**Correspondence:** Claudia Mignani (claudia.mignani@colostate.edu) or Franz Conen (franz.conen@unibas.ch)

**Abstract.** Clouds and precipitation play a critical role in the Earth's water cycle and energy budget. We present ground-level observations of snowfall coinciding with radiosonde launches in Sodankylä, Finland (67.367 °N, 26.629 °E) through a period of eight cold months (October–April) in 2019 and 2020. They comprise 7401 depositing snow particles detected by a snowflake camera and 468 radiosonde profiles. Our results show that precipitating clouds were extending from ground to at least 2.7 km in altitude. Approximately one quarter of them were mixed-phase and the rest were likely fully glaciated. Estimations of the cloud top temperatures indicate that in roughly half of the snowfall events ice might have been initiated through heterogeneous freezing. For such cases, the predicted ice-nucleating particle concentrations active at cloud top temperatures could explain the analysed ice crystal particle concentrations observed near ground in approximately one to two thirds of the cases. For the rest, ice multiplication was likely active. In a warmer climate, the relative proportion of solid to liquid cloud particles will probably decrease, with implications on the radiation balance.

## 1 Introduction

Snowfall has a major impact on the hydrological cycle. It is a requirement for snow cover, which affects the freshwater supply (Barnett et al., 2005) and the albedo of the Earth's surface (Hall and Qu, 2006). Therefore, it is important to understand which atmospheric conditions are driving snowfall. A prerequisite for snowfall is the atmospheric formation of ice crystals. Primary ice crystal formation can be initiated either by homogeneous or heterogeneous freezing, that is by cloud droplets freezing spontaneously below a temperature threshold of $-38$ °C when the relative humidity with respect to ice is above 145% (e.g. Murray et al., 2010) or by ice-nucleating particles (INPs) that promote freezing already at warmer temperatures and lower relative humidity (e.g. Kanji et al., 2017). Once primary ice crystals have formed, other atmospheric ice-related processes may occur, such as ice crystal growth by water vapor deposition, riming, secondary ice formation and aggregation (Fukuta and Takahashi, 1999), followed by deposition and accumulation at the surface, if crystals do not vanish before through sublimation in drier air (Nelson and Baker, 1996). These processes depend on microphysical and dynamical conditions that can lead to

fully glaciated clouds (Costa et al., 2017), i.e. ice clouds. North of the Arctic Circle, warming probably leads to a change from ice clouds to more liquid clouds, whose radiative properties (Shupe and Intrieri, 2004) further accelerate Arctic warming (Tan and Storelvmo, 2019). Clouds containing ice crystals and generating snowfall can be studied for example with space-borne remote sensing techniques (e.g. Liu, 2008; Mülmenstädt et al., 2015; Kikuchi et al., 2017; Casella et al., 2017), which may be combined with ground observations of precipitation (Hanna et al., 2008). Another approach is airborne or ground-based remote sensing (e.g. Delanoë et al., 2013; Kneifel and Moisseev, 2020), sometimes combined with radiosondes (e.g. Silber et al., 2021) and morphological investigation of ice crystals collected in-cloud (e.g. Morrison et al., 2011; Ramelli et al., 2021; Pasquier et al., 2022). Further, radiosondes have been combined with morphological investigations of ice particles collected at ground level in several studies (Power et al., 1964; Jiusto and Weickmann, 1973; Seo et al., 2015).

Pinpointing the origin of even very low-intensity snowfall requires sensitive snowfall detection at ground-level. The simultaneous capture of ice particle habits might enable to disentangle some of the microphysical and dynamical conditions, because ice particles encode atmospheric conditions and duration of growth they were exposed to. The shape, size, degree of riming, and eventually aggregation of ice particles encode the temperature and relative humidity with respect to ice ($RH_{ice}$) and liquid water ($RH_{water}$) at which they grew (Nakaya, 1954). According to the current version of Nakaya's habit diagram (Nakaya, 1954; Magono and Lee, 1966; Bailey and Hallett, 2009), rather large and eventually rimed crystals (including needles, stellar plates, two or three dimensional dendritic crystals, and graupel) are the result of supersaturation with respect to (liquid) water. Below water saturation compact and unrimed ice particles are to be expected (such as plates, columns, radiating plates, or bullet rosettes), which grow very slowly (Kobayashi, 1961). Riming occurs if supercooled water droplets of certain sizes are present (Mossop, 1978) and aggregation of snow particles can be significant at $-17\,°C$ and warmer (Hobbs et al., 1974).

In the Arctic, observations of precipitating clouds were predominantly made at coastal or marine sites (e.g. Fridlind et al., 2012; Delanoë et al., 2013; Mioche et al., 2017; Gierens et al., 2020; Pasquier et al., 2022). Unlike coastal or marine sites, which are associated with high moisture levels, continental regions, such as northern Eurasia, including Northern Finland, are drier and receive less snowfall. There, cloud conditions are likely different from those at coastal or marine sites. Here, we used the approach of combining observations of ice particles at the ground with concurrent radiosonde profiles to identify from what type of clouds, ice or mixed-phase, snowfall predominantly occurs in Northern Finland. Our focus is on identifying general patterns. Therefore, we analyse observations made throughout a total of eight months distributed over two cold seasons.

## 2 Methodology

The experimental setup was located at the Arctic Space Centre of the Finnish Meteorological Institute (FMI) in Sodankylä, in Northern Finland ($67.367\,°N$, $26.629\,°E$; 179 m a.s.l.; Fig. 1).

### 2.1 Description of the site

The site is surrounded by a rural landscape consisting of coniferous forest and swamp (Hirsikko et al., 2014). It is situated just above the Arctic Circle. Due to the Gulf Stream, the climate and vegetation are subarctic, although the stratospheric

meteorology is typically arctic (https://fmiarc.fmi.fi/, last access: 31 January 2022). The site is approximately 500 km from the Atlantic coast in the precipitation shadow of the mountain range along the north coast of Norway with a highest elevation around 1800 m.

In most Scandinavian countries the west coast regions receive more precipitation during the seven colder months (October–April), as compared to the mainland (Fig. 1). In large parts of Finland, including the location of our site, it does not exceed 300 mm. Within Europe, this is comparable to values in large parts of Northern Sweden, Western Russia, Eastern Europe and Eastern Spain.

## 2.2 Snowfall measurements and ice particle image processing

A Multi-Angle Snowflake Camera (MASC, Particle Flux Analytics, USA; Garrett et al., 2012) took automatic photographs of precipitating ice particles from 28 February 2019 to 7 April 2020. During that time, the instrument was continuously in operation, excluding the summer months (May–September), when snowfall is very rare at the site (see monthly weather parameters in Fig. A1) and during a short technical interruption between 11 and 25 October 2019. Slight modifications to the original MASC instrument (e.g. new emitter box, exchanging external plugging connectors and internal wiring, adding a box enclosing the isolating transformer and the computer) were performed before installation to make it run smoothly according to our needs.Since the main components of the instrument, including the cameras, have been kept in their original condition, we expect that our instrument provides the same results as an unmodified MASC. The instrument was installed at two meters above ground next to other field instruments that are part of the World Meteorological Organization (WMO) solid precipitation intercomparison experiment (SPICE; Nitu et al., 2018). Whenever an ice particle falls through the horizontal measurement cross-section of the instrument (roughly 10 cm$^2$), three flashlights and cameras (Unibrain Firewire-800 cameras, image resolution: 5 MP, chip size: 8.8 mm x 6.6 mm, focal length: 12.5 mm), which have a 36-deg angle to each other and an identical focal point, are triggered synchronously. Ice particles are captured against a black background. The images are 2448x2048 pixels, resulting in an optic resolution of ∼33 µm per pixel for particles in focus (Praz et al., 2017).

In the following, we only used images taken by the central camera. Generally, the ice particles that trigger the camera are captured in the middle of the resulting MASC images. MASC images on which the object or a part of it was outside the more or less central area of 1748x1430 pixels were rejected from further analysis. This selected area resulted from the original image by cropping 300 pixels from the top, 318 pixels from the bottom, and 350 pixels from both sides. Ice particles were automatically detected using a function provided by the Open Source Computer Vision library to identify the contours of bright (threshold at 20 out of 256 brightness values) continuous objects. We only considered objects with an area of least 500 continuous pixels, which is equivalent to a projected area of 0.54 mm$^2$, as ice particles suitable for further analysis. Smaller objects could generally not be classified according to criteria useful in the context of this study.

The area, position, maximal length, and maximal width of the projected objects larger than 0.54 mm$^2$ were retrieved from the image. A rectangle encompassing the ice particle was cut from the image and placed, while maintaining the optical size, in the center of a completely black background of a fixed size, i.e. the size of the largest object observed during the overall course of this study. This processing of images made later visual inspection of the large number of mostly small ice particles more

convenient. To ensure fast processing of the data, the files of the MASC images and the pre-defined selection images including
extracted features were collected in an SQLite database.

### 2.2.1    Ice particle classification

Ice particle habits were visually classified. Ice particles with undefinable habits, such as broken off pieces, were classified as
non-specific crystals. Single ice particles with unequivocal shape were divided into nine ice crystal classes (see Fig. 2, i.e.
needles, graupel, spatial dendrites, (planar) dendrites, stellar plates, plates, radiating plates, columns, and rosettes). Aggregated
ice particles were classified as aggregates composed of specific or non-specific single crystals. If aggregates were composed
of a single crystal class, we classified them as such. Only aggregates composed of needles, spatial dendrites, dendrites, stellar
plates, radiating plates, or rosettes were observed (see Fig. A2). Each ice particle with an unequivocal shape was classified into
rimed or unrimed. Unrimed ice particles corresponded to riming degrees of 0 and 1, and rimed particles to riming degrees of 2
to 5 according to Mosimann et al. (1994). Blurred or dark ice particle images were seen as invalid and classified as blurred or
dark, respectively.

## 2.3    Vertical profiles and data processing

As part of the Global Climate Observing System Reference Upper-Air Network (GRUAN; WMO Integrated Global Observing
System station identifier 0-20000-0-02836), vertical profiles of temperature, relative humidity with respect to liquid water,
pressure, wind speed and direction were measured at the site with Vaisala RS41 radiosondes (Jensen et al., 2016). Radiosondes
were launched daily at 11:30 and 23:30 UTC by an automated sonde system (Vaisala AS41 Autosonde system) (Madonna
et al., 2020). The radiosondes were launched on balloons with an ascent velocity of approximately 6 m s$^{-1}$ and data was
recorded between ground level and an average of 28 km in altitude. According to the manufacturer, the humidity sensor of the
radiosondes has a measurement uncertainty of 4% and a response time of $< 0.3$ s for $+20\,°C$ ($< 10$ s for $-40\,°C$), 6 m s$^{-1}$,
and 1000 hPa. For temperature and pressure, the sensor measurement uncertainty above 100 hPa are $0.15\,°C$ (with a response
time $< 0.5$ s) and 0.4 hPa (https://www.vaisala.com/sites/default/files/documents/RS41-SGP-Datasheet-B211444EN.pdf, last
access: 31 January 2022).

We used the GRUAN data product, which provides validated data and meta-data. The vertical profiles of RH$_{ice}$ were calcu-
lated using the saturation vapor pressure ITS90 formulation by B. Hardy (https://www.decatur.de/javascript/dew/resources/it
s90formulas.pdf, last access: 31 January 2022) and averaged with a vertical resolution of 100 m.

## 115   2.4    Ground-based weather parameters

A ground-based weather station provided weather parameters in 10-min time intervals, including temperature, RH$_{water}$, wind
measurements (at 22 m above ground), air pressure, snow depth, cloudiness, horizontal visibility, cloud base height, present
weather, and precipitation determined by different instruments operated by the FMI (https://litdb.fmi.fi/luo0015_data.php,
last access: 31 January 2022). The values for wind speed and direction were average values over 10-minute intervals. The

values for the other parameters were instantaneous values. In the WMO SPICE measurement field, precipitation was measured optically with a Parsivel[2] (OTT HydroMet, UK), which has a measurement accuracy of $\pm$ 20% for solid precipitation between 0.001 mm h$^{-1}$ and 1200 mm h$^{-1}$ (https://www.ott.com/products/meteorological-sensors-26/ott-parsivel2-laser-weather-s ensor-2392/, last access: 31 January 2022), and by weight with a Pluvio[2] (OTT HydroMet, UK), a weighing gauge that has an absolute accuracy of $\pm$ 0.05 mm when averaged over 60 min (https://www.ott.com/products/accessories-109/ott-pluvi

o2-weighing-rain-gauge-963/, last access: 31 January 2022). As a note, both precipitation instruments obtained a reasonable agreement after reprocessing Parsivel[2] raw data (Boudala et al., 2014).

During the period of measurements (i.e. from 28 February 2019 to 7 April 2020, excluding May to September), local air temperature at the site was on average $-5.8\,°C$, with minimum and maximum values of $-34.4\,°C$ and $+16.2\,°C$, respectively. Average relatively humid was 87% and the average wind speed at 22 m above ground (i.e. above tree top) was 2.7 m s$^{-1}$, with a

mean direction from Southwest–WestSouthwest (235 °). Moreover, the ground was covered with snow with a mean snow depth of 70 cm and the total precipitation during the eight months was 295 or 376 mm, depending on the measurement method (optical or by weight, respectively). This amount is not unusual when compared to the local climatology of the period $1970-2020$, as shown in Fig. 1. The average locally-measured height of the lowest cloud base (i.e. vertical visibility) was roughly 1.2 km, and the total cloudiness was determined to be on average 5 oktas. Details of the local monthly weather parameters from February

2019 to April 2020 can be found in Fig. A1.

## 3 Results and discussion

### 3.1 Snowfall events and coinciding vertical atmospheric profiles

We analysed a total of 468 radiosonde profiles that coincided with the MASC instrument being operational (i.e. eight months of two winter seasons). Ice particles captured by the camera and recorded within the 15 minutes prior to each radiosonde

launch were considered as coincident, although the path of the radiosondes is not exactly the same as that of the recorded ice particles. The time span of 15 minutes was chosen for practical reasons so that even during low precipitation intensity several ice particles were recorded. Three coinciding observations were not included in our analysis, as it was either raining (10 Feb 2020 at 11:30 and 23:30 UTC) or snow was likely blowing off the ground during unusually strong winds (30 Mar 2019 at 11:30 UTC). A minimum of 10 images with ice particles of $\geq 0.54$ mm$^2$ in size within these 15-minute intervals before each

radiosonde launch is the criterion for what we consider a snowfall event. Instances with less than 10 such images are considered "no-snowfall".

A snowfall event therefore corresponds to a minimum precipitation intensity of about 0.01 mm h$^{-1}$ (assuming spherical ice particles with 0.4 mm radius). The maximum and median number of images with objects larger than 0.54 mm$^2$ per snowfall event were 630 and 68, respectively (Fig. A3). Consequently, the median concentration of such objects in the precipitating air

column would have been around 76 m$^{-3}$, assuming an average fall speed of 1 m s$^{-1}$ (i.e. Locatelli and Hobbs, 1974). A total of 7401 objects were detected on MASC images coinciding with the radiosonde profiles. More than half of these objects (57%) were blurred, dark, or not entirely visible on the pre-defined image selection. The remaining (3156) were analysed for their

size and visually for ice particle class, riming degree, and aggregation. These analysed objects had mean and median projected areas of 1.8 mm$^2$ and 1.1 mm$^2$, respectively (Fig. 3).

Out of 468 radiosonde profiles coinciding with the MASC instrument being operational, 62 were classified as snowfall events and the rest were considered no-snowfall (Fig. A3). For each 15-min event, ground-based meteorological parameters were calculated from two weighted 10-min resolved data points and summarised (Fig. 4). During snowfall, the median and maximum wind speeds at 22 m above ground were 3.0 m s$^{-1}$ and 5.6 m s$^{-1}$, respectively, and the median air temperature was $-3.1\,°C$. This is consistent with the results of Kochendorfer et al. (2017) for precipitation periods in the winter seasons of the

time period $2013-2015$. Moreover, during snowfall events, the median wind at 22 m above ground was from the East-Southeast $(110\,°)$ although at greater altitude (2.7 km) it was mostly from West to Southwest (Fig. A4) and the median precipitation rate was relatively low with 0.20 mm h$^{-1}$, as determined optically with the Parsivel$^2$. Somewhat higher values were measured with the Pluvio$^2$ (0.32 mm h$^{-1}$). During snowfall, the horizontal visibility was on average 2693 m, the average base height (or vertical visibility) of the lowest cloud was 285 m, the sky was fully cloud covered (cloud cover = 8) and it was slightly snowing (code of

"present weather" = 71). This suggests that snowfall was produced by probably stratiform clouds almost touching ground level. For events without snowfall, the median precipitation rate was zero, regardless of the measurement technique. The comparison of various other meteorological parameters between events with and without snowfall showed that some parameters differed in terms of range and median (i.e. RH$_{water}$, pressure, wind direction, horizontal visibility, first cloud base height, cloudiness, and present weather). Other parameters were relatively similar during snowfall as during no-snowfall events (i.e. air temperature,

snow depths, and wind speed).

The RH$_{ice}$ of the profiles during snowfall showed a consistent pattern (Fig. 5a). RH$_{ice}$ was close to saturation at ground, increased slightly within the first few hundred meters, and remained above and close to 100% RH$_{ice}$ in the lower troposphere up to roughly 3 km. Above this altitude, RH$_{ice}$ values were more scattered among the profiles, with some profiles showing supersaturation with respect to ice up to 10 km. Consistently during snowfall, RH$_{water}$ showed values close to 100% from

ground to a few hundreds of meters, followed by a slight decrease with increasing altitude up to 3 km, and a stronger variation above this altitude (Fig. A5a). RH$_{ice}$ and RH$_{water}$ of the profiles not associated with solid precipitation showed no such consistent pattern (Fig. 5b and Fig. A5b).

## 3.2  Necessary or sufficient RH$_{ice}$ conditions for snowfall

To test whether RH$_{ice}$ close to saturation throughout the lower $\sim$ 3 km of the atmosphere is a necessary or sufficient con-

dition for snowfall at the site, we evaluated a range of RH$_{ice}$ values and altitude ranges using different metrics, including accuracy (ACC), critical success index (CSI), and Heidke skill score (HSS), as shown in Fig. 6. An overview of the indices can be found in the Appendix (Table A1). All three indices have an optimal value of 1, favour hits, and penalise both misses and false alarms. Based on the results of three indices, the best scores were obtained for the following combination of values (from hereon: criterion): running mean RH$_{ice}$ $\geq$ 97% throughout the lower 2.7 km of the atmosphere. Choosing different values for

running mean RH$_{ice}$ and altitude reduced the overall accuracy of the prediction of snowfall events, although a running mean RH$_{ice}$ $\geq$ 96% throughout the lower 2.7 km of the atmosphere also yielded a very good score.

Of the total of 468 radiosonde profiles, 64 met the criterion of a running mean $RH_{ice} \geq 97\%$ throughout the lower 2.7 km (Fig. 7). In 81% of them (52 of 64) snowfall was observed (*hits*). In 10 cases the criterion was not met but snowfall was observed (*misses*). In 12 cases the criterion was met but no snowfall was observed (*false alarms*) and in 394 cases the criterion was not met and no snowfall was observed (*correct negatives*). In other words, these specific thresholds of $RH_{ice}$ and altitude correctly predicted 84% (52 of 62) and 97% (394 of 406) of the cases with and without snowfall, respectively. Although the criterion of a running mean $RH_{ice} \geq 97\%$ throughout the lower 2.7 km was neither absolutely necessary nor sufficient for snowfall to occur, it still separates the majority of snowfall events from no-snowfall events and vice versa.

## 3.3 Snowfall rate and type

In this section we will focus on the 52 snowfall events coinciding with a running mean $RH_{ice} \geq 97\%$ throughout the lower 2.7 km (i.e. *hits*, Fig. 7a), which seem to represent the predominant condition in which snowfall occurs in Northern Finland (dates of the events are listed in Table A2). These events were associated with optically measured median and maximum snowfall rates of 0.23 mm h$^{-1}$ and 1.37 mm h$^{-1}$, respectively, as derived from the Parsivel$^2$ precipitation measurements. Only three of them were associated with snowfall rates <0.05 mm h$^{-1}$ (Fig. A7 top left panel). In general, the snowfall rates for the 52 snowfall events we now consider were significantly higher than for the other cases (i.e. *misses*, *false alarms* and *correct negatives*), as shown in Fig. A7.

Excluding the invalid objects (i.e. blurred or dark, Fig. A2), a total of 2853 analysed ice particles were captured during these 52 snowfall events. They had a median projected area of around 1.1 mm$^2$ (not shown) and a very similar size distribution to that of all events, which are shown in Fig. 3. Many of these objects on the visually inspected images had undefinable habits (Fig. A2). Only 29% of them (827 of 2835) could unequivocally be classified by their habit. Nine different unequivocal crystal habits were observed (Fig. 2). The majority of them were single ice crystals and unrimed (Fig. 8). Most ice crystals had habits that form below liquid water saturation (i.e. single plates, columns, radiating plates, rosettes, or their aggregates). Notably, the shapes that grow below the liquid water saturation were often unrimed (not shown). The most common habit of the classifiable ice particles was radiating plates, with a share of 54% (447 of 827).

## 3.4 Properties of ice crystals and air masses related to maximum $RH_{water}$ along the profile

We grouped the 52 snowfall events under consideration according to their maximum running mean $RH_{water}$ along the radiosonde profiles into three similarly large sub-groups: < 98%, $\geq$ 98% and < 99%, and $\geq$ 99%, as shown in Fig. 9a-c. For each sub-group, we determined the corresponding ice particle size distribution and ice particle classes (Fig. 9d-i). Overall, 14 of 52 events (27%) reached a maximum running mean $RH_{water}$ along the profile between 99% and 100% (Fig. 9c). These events coincided with ice particles that were (1) larger (Fig. 9f), (2) more often rimed and aggregated, and (3) more often associated with habits that grow above the water saturation line (i.e. needles, graupel, spatial dendrites, dendrites, and stellar plates; Fig. 9i) as compared to those events (38 of 52) that did not reach a maximum running mean $RH_{water}$ of 99% (Fig. 9a,b,d,e,g,h). Hence, only around one quarter of all precipitating clouds seem to have been mixed-phase and the rest were probably fully glaciated. The lowest proportion of riming and aggregation was found for events for which the maximum running mean $RH_{water}$ did not

exceed 98% (Fig. 9a,d,g). These results suggest that the humidity measurements by the radiosondes during our measurement campaign and in the lower 3 km of the atmosphere were more accurate near saturation with respect to water than the 4% promised by the manufacturer, at least when averaged over 500 m.

Most air masses associated with the 52 snowfall events tended to come from a southerly to westerly direction and crossed marine areas shortly before arriving at the site (Fig. 10 and Fig. A4b), which suggests that the Baltic Sea, the Norwegian Sea and the North Sea were the major sources of moisture for snowfall in the observed cases. The few air masses that did not pass over marine areas recently, but crossed Russia and Eastern Europe, were associated with maximum running mean $RH_{water}$ below 99%.

### 3.5 Likely underlying ice formation processes

To determine whether heterogeneous or homogeneous freezing might have triggered the initial ice formation during the considered 52 snowfall events (coinciding with running mean $RH_{ice} \geq 97\%$ throughout the lower 2.7 km), we extracted from the radiosonde profiles the likely cloud top temperature, which is the coldest temperature at which initial ice might have formed. Since a higher level cloud could seed a lower level cloud (e.g. Vassel et al., 2019; Ramelli et al., 2021; Proske et al., 2021), we determined the cloud top of the highest possible seeding cloud. Here we define successive values of the running mean from five consecutive 100 m averaged $RH_{ice}$ with increasing height $\geq 100\%$ as a cloud. For each cloud, we determined the cloud base height, cloud top height, and cloud top temperature. The cloud top temperature is the minimum temperature between the cloud base and cloud top measured by the radiosonde temperature sensor. When two clouds were on top of each other, we further determined the distance between the cloud top height of the lower-level cloud and the cloud base height of the upper-level cloud. If this distance was below a certain threshold (0.2, 0.5, or 1 km), we considered the two clouds as potential seeder-feeder clouds. If there were more than two clouds on top of each other, we determined the highest seeder cloud for which the distance threshold with increasing height holds. Finally, we determined the cloud top temperature of the highest seeder cloud for each case (also described as cloud top temperature from here on). Hence, we take into account that up to a certain (vertical) distance between clouds, ice crystals from the upper cloud (seeder) could fall through the unsaturated layer without fully sublimating, thus seeding the lower cloud (feeder). It is noteworthy that our threshold for seeder-feeder consideration is based only on the distance between clouds. However, whether an ice crystal fully sublimates between two clouds depends on several factors such as the crystal's size and habit when entering unsaturated conditions or by the degree of unsaturated air. In the absence of in-cloud crystal information, we cannot make detailed calculations. To compensate for this uncertainty, we show cloud top temperature estimates for three different distance thresholds. Note that we do not distinguish between cases with a single cloud and those with seeder-feeder clouds. Depending on the assumed distance threshold between potential seeder and feeder clouds, between 38% and 58% of the cloud tops were colder than $-38\,°C$ (Fig. 11a). Here, freezing would have been initiated via homogeneous freezing (Murray et al., 2010). However, between 42% and 62% of the cloud tops may have had temperatures warmer than $-38\,°C$. In these cases, ice formation would have been initiated via heterogeneous freezing. Since $RH_{ice}$ and $RH_{water}$ along the profiles were often $< 130\%$ and $< 100\%$, respectively (i.e. Figs. 5a and 9a-c), we speculate that

initial ice formed via pore condensation and freezing onto mineral dust, biogenic particles or other INPs (Kanji and Abbatt, 2006; David et al., 2019).

We may estimate whether enough INPs were present to explain the number of ice particles observed by using the empirical parameterisation by Schneider et al. (2021):

$$INP\left(T\right) = 0.1 \cdot exp\left(a1 \cdot T_{amb} + a2\right) \cdot exp\left(b1 \cdot T + b2\right) stdL^{-1} \tag{1}$$

where $INP\left(T\right)$ is the number concentration of INPs (in stdL$^{-1}$) active at temperature $T$ (in K), $T_{amb}$ is the ambient air temperature at ground level (in K), and $a1$, $a2$, $b1$, and $b2$ are empirically fitted parameters ($a1 = 0.074$ K$^{-1}$, $a2 = -18$, $b1 = -0.504$ K$^{-1}$, $b2 = 127$). We used daily averaged local temperatures for $T_{amb}$ to calculate the INP number concentration active at cloud top temperatures. Considering the cases with cloud top temperatures warmer than $-38\,°C$, the median INP concentrations were 15 m$^{-3}$, 28 m$^{-3}$, and 410 m$^{-3}$, respectively, depending on the assumed distance between potential seeder and feeder clouds (Fig. 11b). Since the necessary aerosol properties were not measured at the site at the time period of interest, it was not possible to use other existing INP parameterisations (e.g. DeMott et al., 2015; Ullrich et al., 2017) to qualitatively evaluate the associated uncertainty. However, the here predicted INP concentrations are similar to the range of observed long-term median INP concentrations active between $-20\,°C$ (60 m$^{-3}$) and $-30\,°C$ (690 m$^{-3}$) in condensation freezing mode (i.e. RH$_{water} = 101\%$) at an Arctic site in Svalbard (474 m a.s.l.; Schrod et al., 2020).

Finally, for cases with cloud top temperatures warmer than $-38\,°C$, we estimate the likely ice multiplication factor, which is the observed snowflake concentration divided by the estimated INP concentration (Fig. 12). For a distance between potential seeder and feeder clouds of 1 km the median ice multiplication factor was less than 1, indicating that the median number of estimated INPs would have been sufficient to generate the median number of observed ice particles. For smaller thresholds (0.2 and 0.5 km), median ice multiplication factors of 1.8 and 3 would need to be invoked to explain the median observations. Also, we find higher median ice multiplication factors for cases that are mixed-phase clouds compared to those that are likely ice clouds. A closer look at the individual events shows that in 36% to 66% of the cases the ice multiplication factor was higher than 1, depending on the threshold to determine the highest seeder cloud. Therefore, secondary ice formation processes were probably active in one to two thirds of the cases (where ice formation was initiated through heterogeneous freezing). Highest ice multiplication factors were found for cloud top temperatures between $-3\,°C$ and $-10\,°C$, ranging from 10 to 1000. This could be indicative of rime-splintering (Hallett and Mossop, 1974). For temperatures between $-10\,°C$ and $-20\,°C$, the multiplication factors reached values up to 10, which could be indicative of ice multiplication from ice–ice collision of dendrites followed by breakup (Vardiman, 1978; Mignani et al., 2019). This secondary ice mechanism was shown to be linked to the collision force and the riming degree, with a number of observed fragments per collision below 1 for unrimed dendrites and below 8 for lightly rimed dendrites (Vardiman, 1978; Phillips et al., 2017). Note that, we saw some broken-off branches of dendrites (Fig. A2), suggesting at least occasional ice–ice collision followed by breakup was active. Other ice multiplication processes exist (Korolev and Leisner, 2020) and could be active. For cloud top temperatures below $-20\,°C$, sufficient INPs were likely active to explain the observed number of snowflakes. In general, the ice multiplication factors in relation to the temperatures

observed are consistent with previous observations (see Fig. 14 in Wieder et al., 2022) and show that ice multiplication played a role in the vast majority of the cases associated with cloud top temperatures $\geq -15\,°C$.

## 4    Conclusions

Overall, about 70% of all classifiable ice crystals observed during the 52 snowfall events (i.e. events that had a minimum of 10 images with ice particles of $\geq 0.54\ \mathrm{mm^2}$ in size within 15-minute intervals) that were correctly predicted using our basic $\mathrm{RH_{ice}}$ criteria (i.e. running mean $\mathrm{RH_{ice}} \geq 97\%$ throughout the lower 2.7 km) in Northern Finland had habits that form below the water saturation. In a similar proportion of radiosonde profiles, the running mean $\mathrm{RH_{water}}$ did not reach values $\geq 99\%$. This suggests that snowfall in this region was derived from ice clouds in nearly three quarters of the cases. As ice particles grow very slowly in such conditions, the lower atmosphere can not be much below saturation with respect to ice for snowfall to occur. Otherwise, the small, often unrimed ice particles would not reach the surface before completely sublimating. These precipitating clouds are probably stratiform clouds that extend from close to the ground to at least 2.7 km altitude. At that altitude the air masses arrived mostly from West and Southwest. Probably, the Baltic Sea, the Norwegian Sea, and the North Sea were the major source of moisture generating snowfall in Northern Finland. Due to orographic lifting at the coastal mountain range, North Atlantic air masses might already have lost much of their initial moisture content when arriving in Northern Finland.

In a warmer climate, the partitioning of ice particles and liquid cloud droplets in Arctic clouds may shift in favour of liquid droplets. Therefore, a greater proportion of mixed-phase clouds than currently observed can be expected. This could lead to snowfall events with a greater proportion of larger snowflakes, rimed ice particles, and such crystal habits that grow above water saturation compared to today. Rime-splintering and other secondary ice formation processes requiring liquid droplets could become more frequent, which would likely increase the ice multiplication factor. Such cloud microphysical changes depend on the availability of INPs, which are currently responsible for perhaps half of the initial ice formation during snowfall in Sodankylä. In the future, the effects of eventual changes in predominant cloud phase should be further investigated through studying cloud and precipitation microphysics as well as cloud radiation properties to better understand potential feedback mechanisms between climate change and cloud properties.

*Data availability.*    The RS41 GRUAN data products are available via https://www.gruan.org/instruments/radiosondes/sonde-models/vaisala-rs41 (last access: 31 January 2022). The ground-based weather parameters are accessible via https://litdb.fmi.fi/luo0015_data.php (last access: 31 January 2022). The MASC images are available from the authors upon request.

*Author contributions.*    CM and FC conceived the study. LZ modified, installed and maintained the MASC instrument with help from FC and know-how from AB. RK performed the radiosonde launches. LZ processed the MASC images and CM classified the ice particles. CM performed the analysis and prepared the figures. CM, LZ, RK, AB and FC interpreted the data. CM drafted the manuscript with contributions from all co-authors.

*Competing interests.* The authors declare that they have no conflict of interest.

*Acknowledgements.* The authors are grateful to the Finnish Meteorological Institute for providing the infrastructure as well as the ground-based weather parameters on site. We thank the team in Sodankylä for their support on site. Special thanks to Jaakko Siltakoski for providing images of the MASC instrument and helping during the installations in the SPICE field. We greatly appreciate the support of Antti Poikonen
for establishing remote access to the MASC and Michael Sommer for his help in downloading the GRUAN data. We are grateful to Pedro Batista for his advice during coding in R and his help in creating the maps in GIS. We acknowledge the NOAA Air Resources Laboratory for providing access to the HYSPLIT model through the READY website (https://www.ready.noaa.gov/HYSPLIT.php, last access: 31 January 2022). We acknowledge financial support by the Swiss National Science Foundation as well as the Department of Environmental Sciences at University of Basel. We express our gratitude to the two anonymous referees for their constructive comments.

*Financial support.* This study has been supported by Swiss National Science Foundation (grant number 200021_169620) and the Department of Environmental Sciences, University of Basel.

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

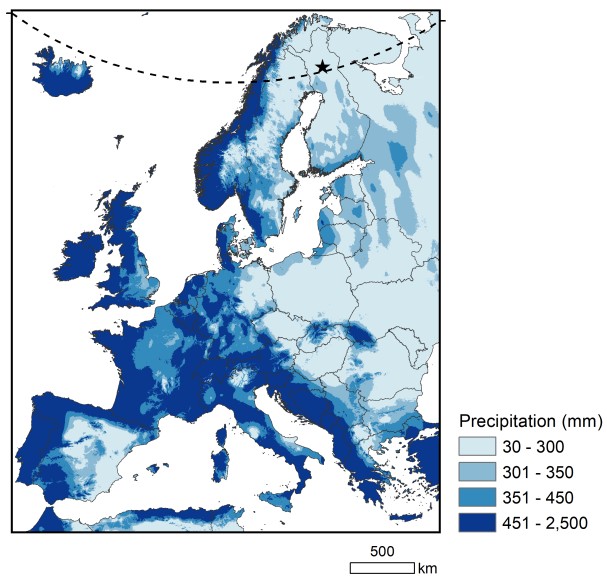

**Figure 1.** Overview of the geographical location of the experiment setup in Sodankylä, Finland (black star), which is just above the Arctic Circle (dashed line). The average seven-month total precipitation (in mm) for the colder months per year (January-April and October-December) in the period 1970−2000 are shown in the map derived from the monthly climate data from WorldClim (Fick and Hijmans, 2017, see also https://www.worldclim.org/data/worldclim21.html, last access: 31 January 2022).

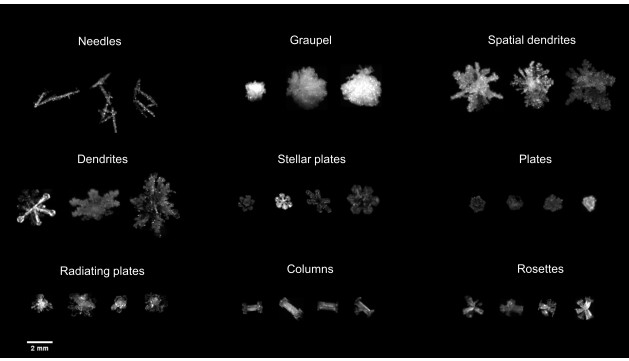

**Figure 2.** Some examples of single, specifiable ice particles captured by the MASC and their shape. Needles were only captured as aggregates.

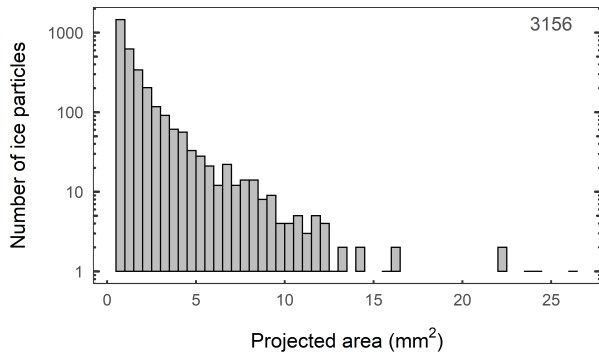

**Figure 3.** The number of analysed ice particles (n = 3156) by intervals of projected area (mm$^2$) roughly followed a power function. Analysed ice particles captured by the camera during the 15-min intervals prior to the radiosonde launches throughout the whole time span of interest (28 February 2019 to 7 April 2020) were considered.

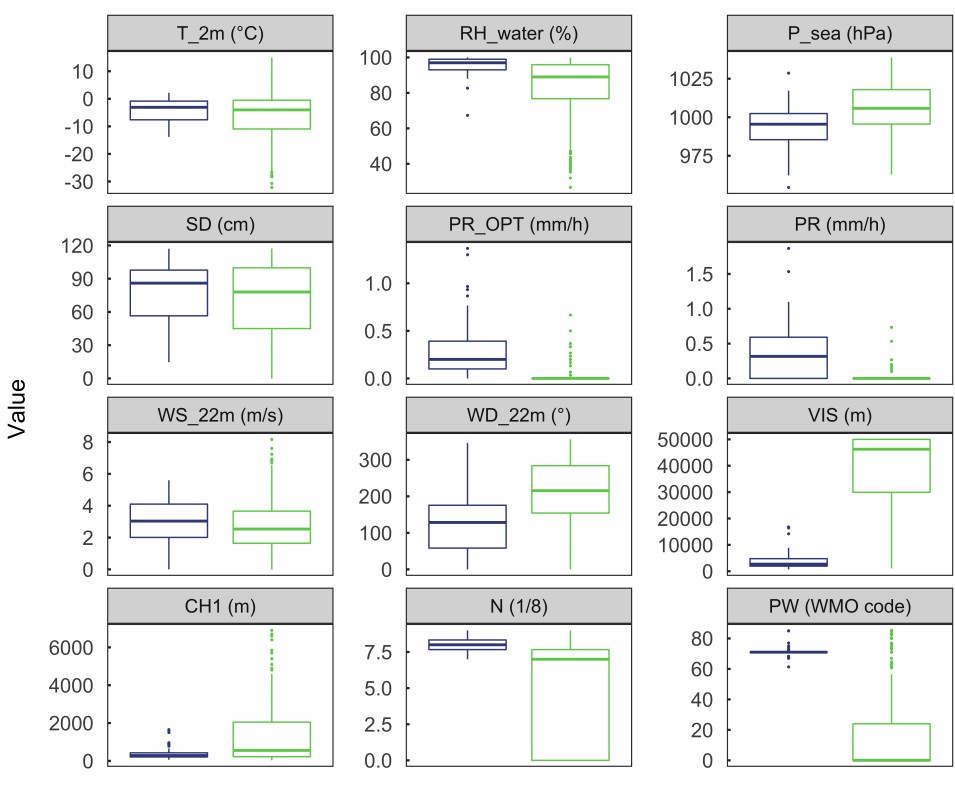

**Figure 4.** Boxplots (i.e. median (thick line), interquartile range (box), minima and maxima (whiskers), outliers (dots)) of weather parameters measured at the site coinciding with radiosonde profiles during events considered as snowfall (n = 62, blue) and as no-snowfall (n = 406, green). The following parameters are shown from the upper left to the bottom right: ambient air temperature at 2 m above ground (T_2m in °C), the relative humidity with respect to water (RH_water in %), the pressure at sea level (P_sea in hPa), the snow depth (SD in cm), the precipitation rate measured optically (PR_OPT in mm h$^{-1}$) and by weight (PR mm h$^{-1}$), the wind speed at 22 m above ground (WS_22m in m s$^{-1}$), the wind direction at 22 m above ground (WD_22m in °), the horizontal visibility (VIS in m), the height of the lowest cloud base (CH1 in m), the total cloudiness (N, a number between 1 (cloud free) and 8 (cloud covered)), and the present weather (PW using the WMO Code 4680).

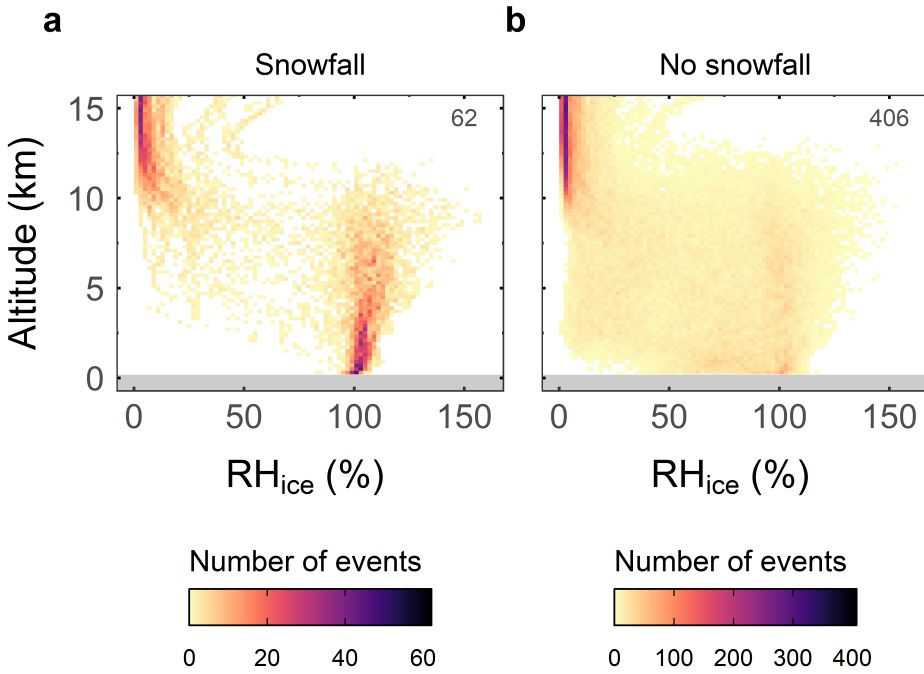

**Figure 5.** Relative humidity with respect to ice ($RH_{ice}$) within intervals of 200 m height and 2% $RH_{ice}$ retrieved from the radiosonde profiles during (**a**) snowfall and (**b**) no-snowfall events. The total number of events for each group is provided in the upper right corner of each panel. The color scale ranges from zero to the total number of events for each group, so the color scale for each panel is different.

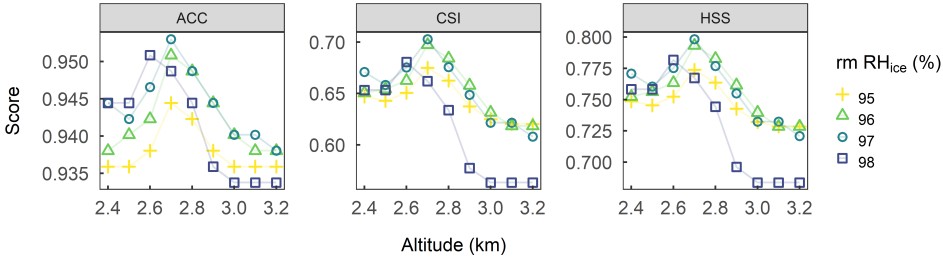

**Figure 6.** Scores to assess the prediction of snowfall using the running mean $RH_{ice}$ values and altitude ranges were best for running mean $RH_{ice} \geq 97\%$ throughout the lower 2.7 km above ground. A set of different lower threshold values was investigated ranging for a running mean $RH_{ice}$ from $\geq 95\%$ to $\geq 98\%$ throughout altitude ranges from ground level to between 2.4 km and 3.2 km above ground. Running means in $RH_{ice}$ were calculated of five consecutive 100 m layers and the running mean was allocated to the lowest of the five layers. The following scores are shown: accuracy (ACC), critical success index (CSI), and Heidke skill score (HSS). See Table A1 for more information on the metrics.

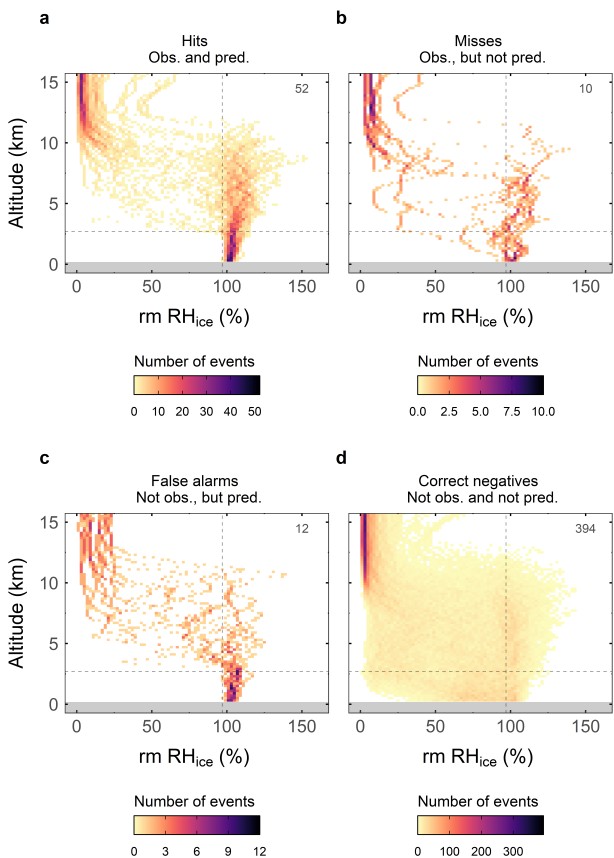

**Figure 7.** Observed values of running mean RH$_{ice}$ binned in 200 m intervals and 2% steps of RH$_{ice}$ for (**a**) *hits*, (**b**) *misses*, (**c**) *false alarms*, and (**d**) *correct negatives*. Snowfall was predicted for cases when running mean RH$_{ice}$ measured along the radiosonde profile was $\geq 97\%$ (vertical dashed line) throughout the lower 2.7 km (horizontal dashed line) of the atmosphere. The total number of events per group is shown in the upper right corner of each panel. The color scale ranges from zero to the total number of events for each group, so the color scale for each panel is different.

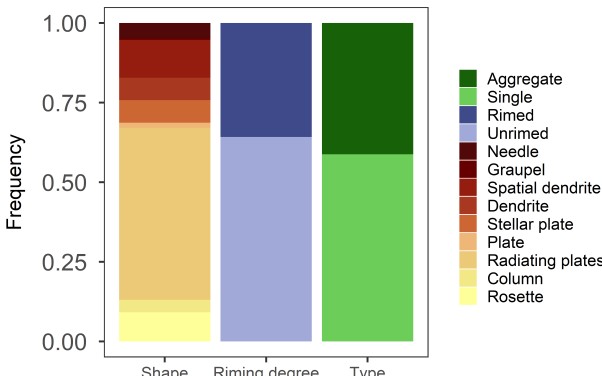

**Figure 8.** Analysis of (827) unequivocal ice particles that were captured during the 52 snowfall events coinciding with a running mean $RH_{ice}$ $\geq 97\%$ throughout the lower 2.7 km. Frequency with respect to shape (i.e. as shown in Fig. 2), riming degree (i.e. unrimed or rimed) and type (i.e. single or aggregate) are shown.

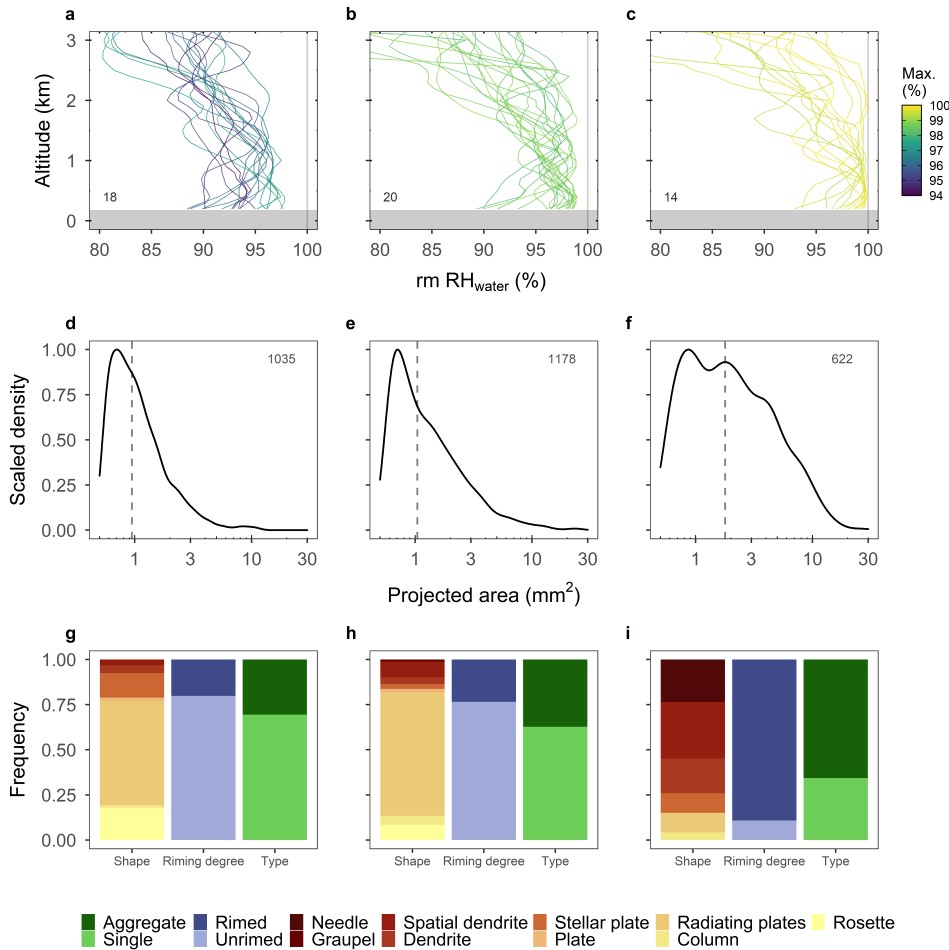

**Figure 9.** (**a, b, c**) Running mean RH$_{water}$ profiles coinciding with snowfall and running mean RH$_{ice}$ ≥ 97% throughout the lower 2.7 km (52 events) grouped by the maximum running mean RH$_{water}$. The maximum running mean RH$_{water}$ along the altitude profile is shown in colour. (**a**) Profiles with a maximum running mean RH$_{water}$ < 98%. (**b**) Profiles with a maximum running mean RH$_{water}$ ≥ 98% and < 99%. (**c**) Profiles with a maximum running mean RH$_{water}$ ≥ 99%. For each panel, the number of profiles is indicated in the left bottom corner. (**d, e, f**) The scaled density of ice particles by projected area (mm$^2$). All the analysed ice particles of the profiles in the above lying panel were considered. The dashed vertical line indicates the median area. The number of analysed ice particles is indicated by the number provided in the upper right corner. During the 52 snowfall events, a total of 2835 ice particles were analysed. (**g, h, i**) Same as Fig. 8, but considering only the specifiable ice particles coinciding with the events of the above lying panels.

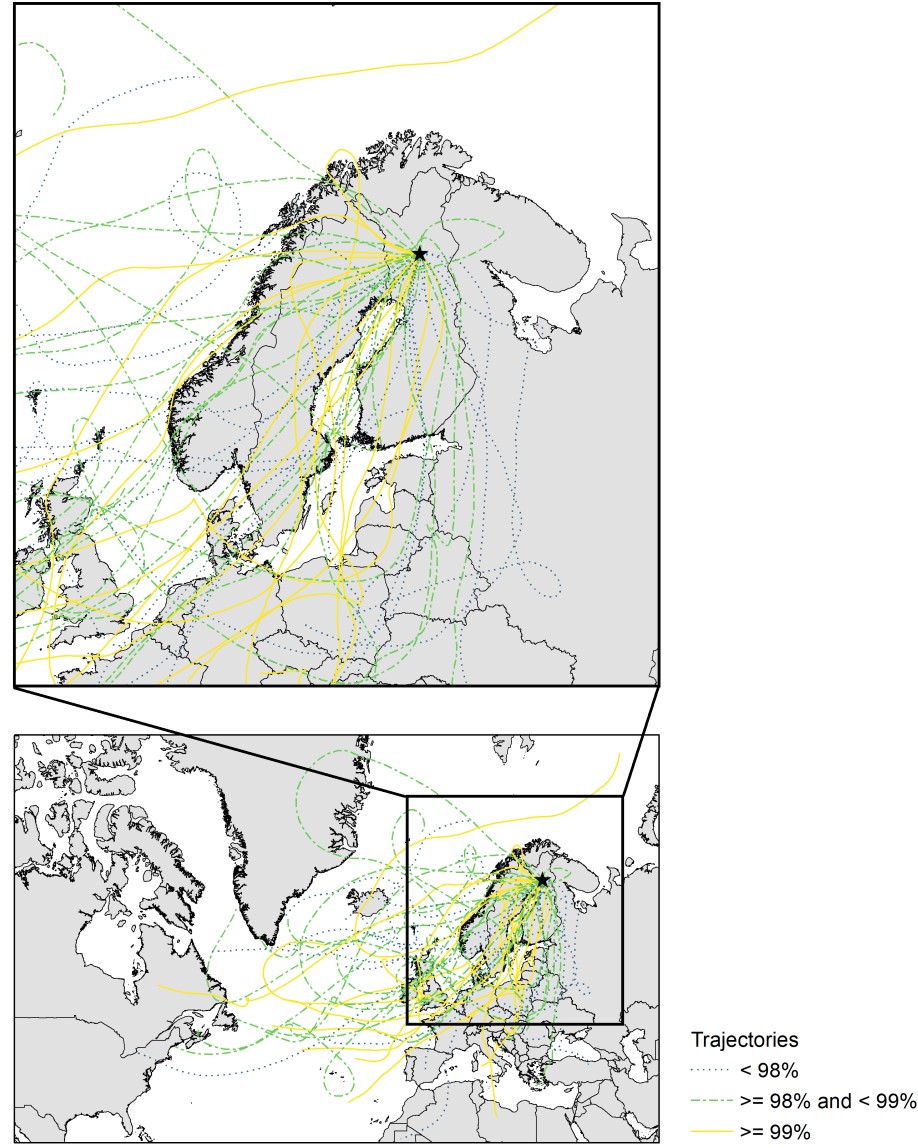

**Figure 10.** Five-day backward trajectories (lines) arriving at the measurement site in Sodankylä, Finland (black star) at 2.7 km above ground at full hour (i.e. 11 or 23 UTC) for each of the 52 snowfall events. The group of maximum running mean RH$_{water}$ (as in Fig. 9) is shown in colour (< 98%, dotted blue; $\geq$ 98% and < 99%, dashed green; < 99% continuous yellow). An inset map is shown in the upper panel. The trajectories were calculated using the NOAA's Hybrid Single-Particle Lagrangian Integrated Trajectory (HYSPLIT) model. One trajectory per event was computed using the GDAS one-degree meteorology dataset and the "model vertical velocity" vertical motion method.

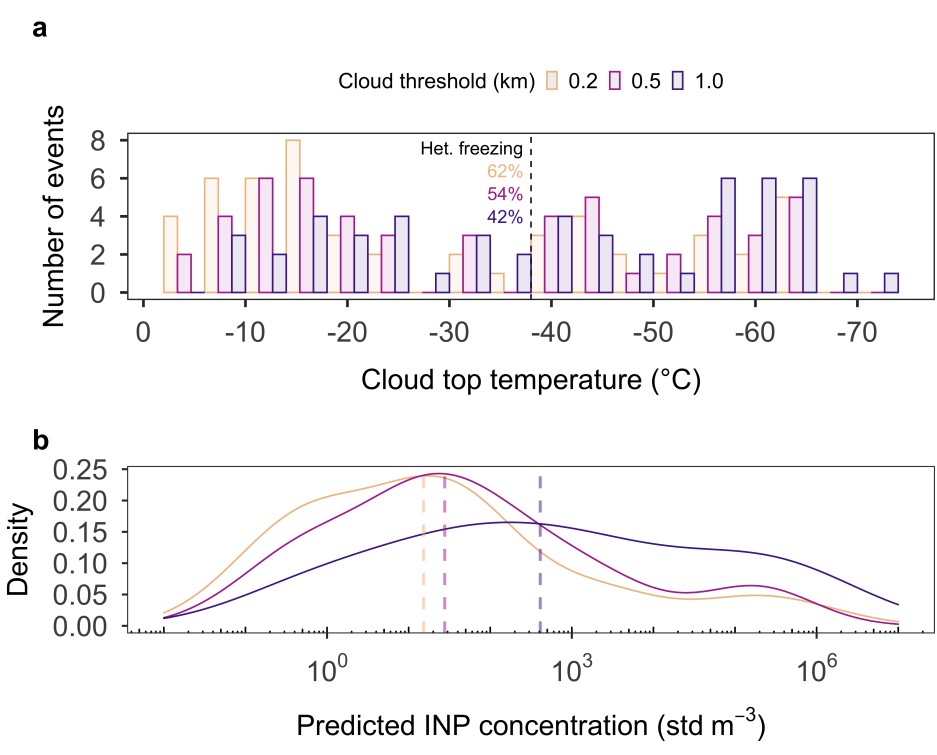

**Figure 11.** (**a**) The cloud top temperatures of the highest possible seeder-feeder cloud in $4\,°C$ intervals of the 52 events coinciding with running mean $RH_{ice} \geq 97\%$ throughout the lower 2.7 km. The cloud top temperature was derived using the radiosonde measurements which is described in detail in Sect. 3.5. We used the following thresholds for the distance between clouds to be considered as seeder-feeder clouds: $\leq 0.2$ km (orange), $\leq 0.5$ km (pink), and $\leq 1.0$ km (purple). The fraction of events with cloud top temperatures above $-38\,°C$ is given in percent next to the dashed line. This is an estimation of the fraction of events for which the first ice crystals were likely formed via heterogeneous freezing. (**b**) Density of the INP concentration for the fraction of events with cloud top temperatures above $-38\,°C$, using the different thresholds to account for seeder-feeder clouds (in color) as shown in panel a. The respective median concentrations are shown by the dashed lines.

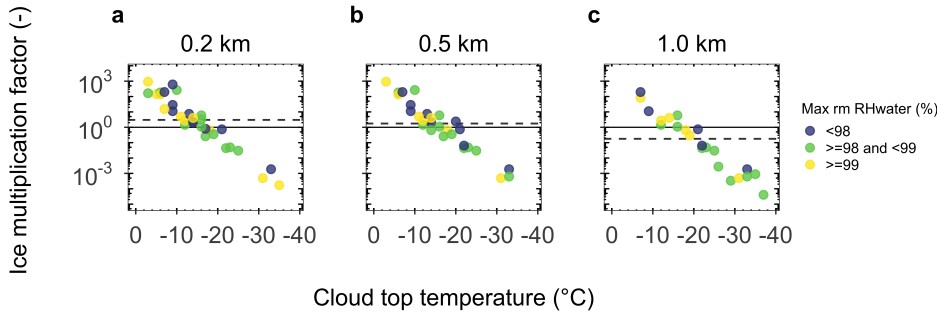

**Figure 12.** The ice multiplication factor versus cloud top temperature (°C) for each of the 52 snowfall events (coinciding with running mean $RH_{ice} \geq 97\%$ throughout the lower 2.7 km) that were associated with cloud top temperatures warmer than $-38\,°C$ determining the highest possible seeder-feeder cloud using a distance threshold of (**a**) 0.2 km, (**b**) 0.5 km, and (**c**) 1.0 km. The colors are indicative of the associated maximum running mean $RH_{water}$ ($< 98\%$, blue; $\geq 98\%$ and $< 99\%$, green; $\geq 99\%$, yellow). The median ice multiplication factor is shown by the dashed line. The solid line is drawn at an ice multiplication factor of 1.

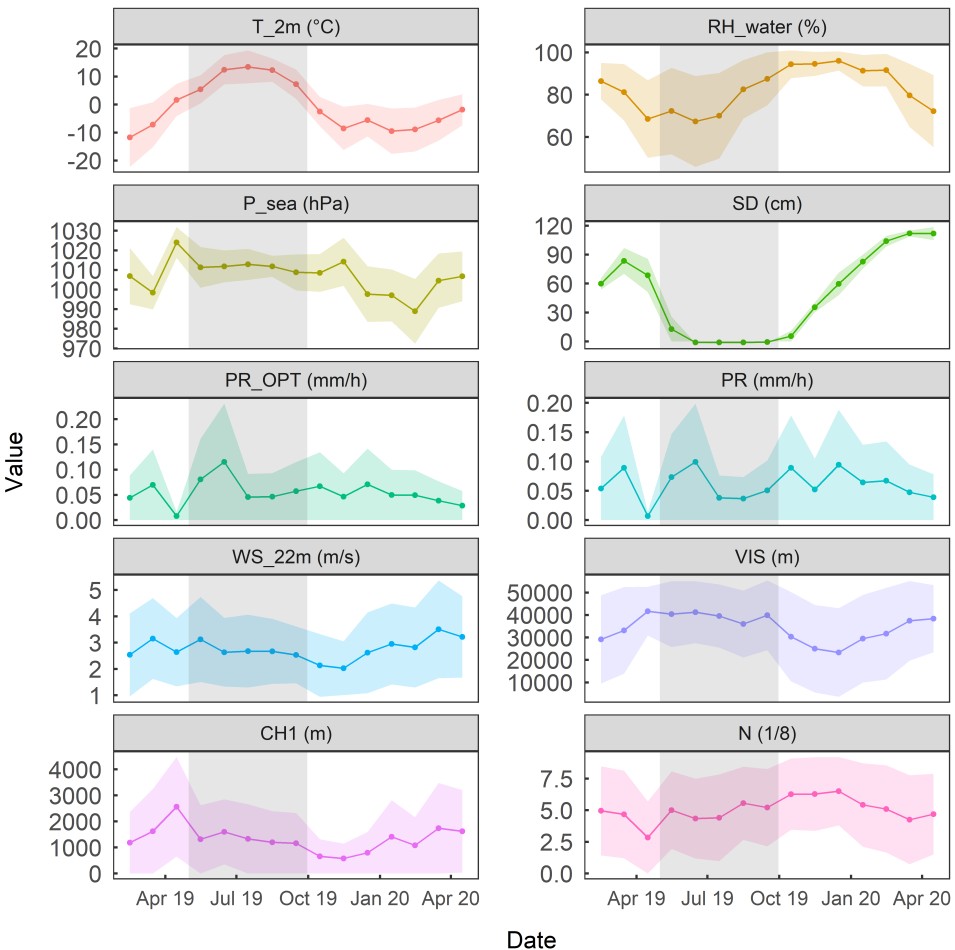

**Figure A1.** Time series of monthly weather parameters (means (dots and connecting lines) and standard deviations (ribbons)) at the measurement site in Sodankylä from February 2019 to April 2020. The following variables are shown (from the upper left to the bottom right): ambient air temperature at 2 m above ground (T_2m in °C), relative humidity with respect to water (RH_water in %), pressure at sea level (P_sea in hPa), snow depth (SD in cm), precipitation rate measured optically (PR_OPT in mm h[-1]) and by weight (PR mm h[-1]), wind speed (WS at 22 m above ground in m s[-1]), horizontal visibility (VIS in m), height of the lowest cloud base (CH1 in m), and total cloudiness (N, a value between 1 and 8). The grey areas mark the months from May to September, that were excluded from our analysis.

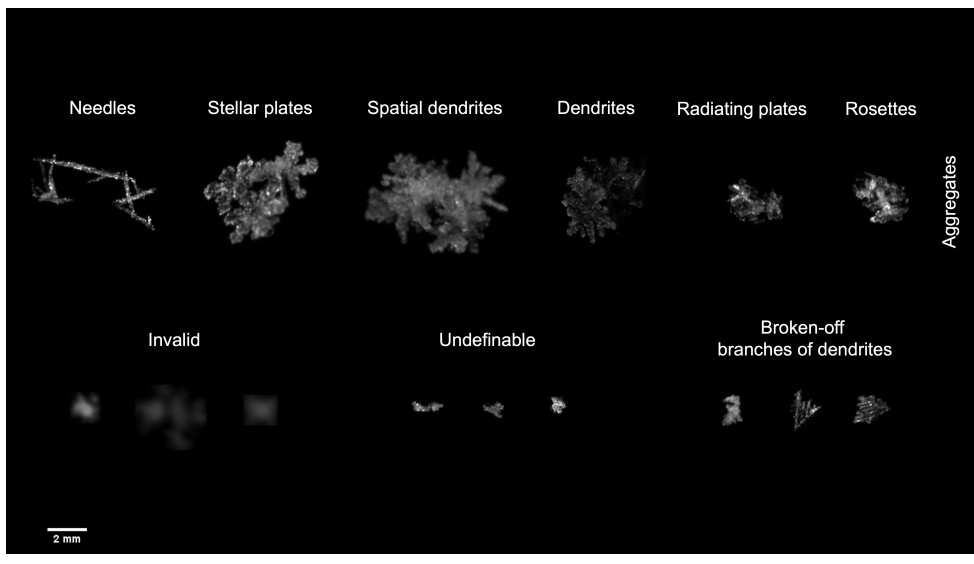

**Figure A2.** Similar to Fig. 2, but with further examples of images captured by the MASC. Aggregates of specifiable ice particle shapes are arranged in the top row. Invalid and undefinable particle shapes as well as broken-off branches of dendrites are shown in the bottom row.

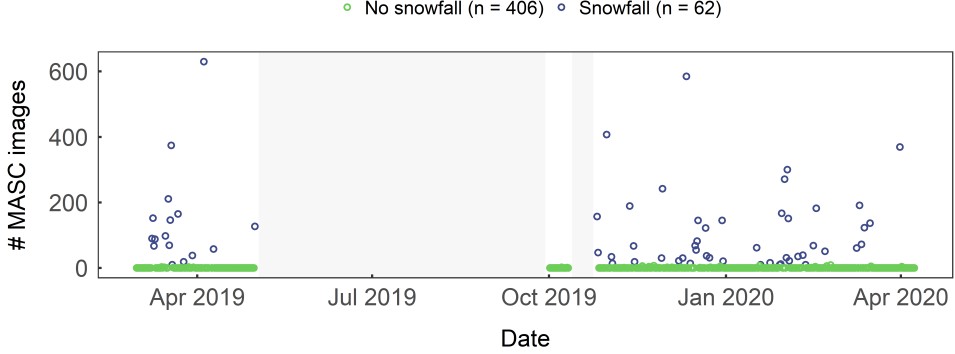

**Figure A3.** The number of images with objects larger than 0.54 mm$^2$ captured by the operational MASC instrument during coinciding radiosonde profiles from 28 February 2019 to 7 April 2020. Radiosondes were launched twice a day, at 11:30 and 23:30 UTC. Images considered were captured within a time span of 15 minutes before each radiosonde launch. Cases (i.e. 15-min time intervals) with 10 or more of such images are classified as snowfall events (n = 62, blue dots). The others are classified as no-snowfall events (n = 406, green dots). No data (gray area) was collected in the summer months (May to September) and during a technical interruption between 11 and 25 October 2019.

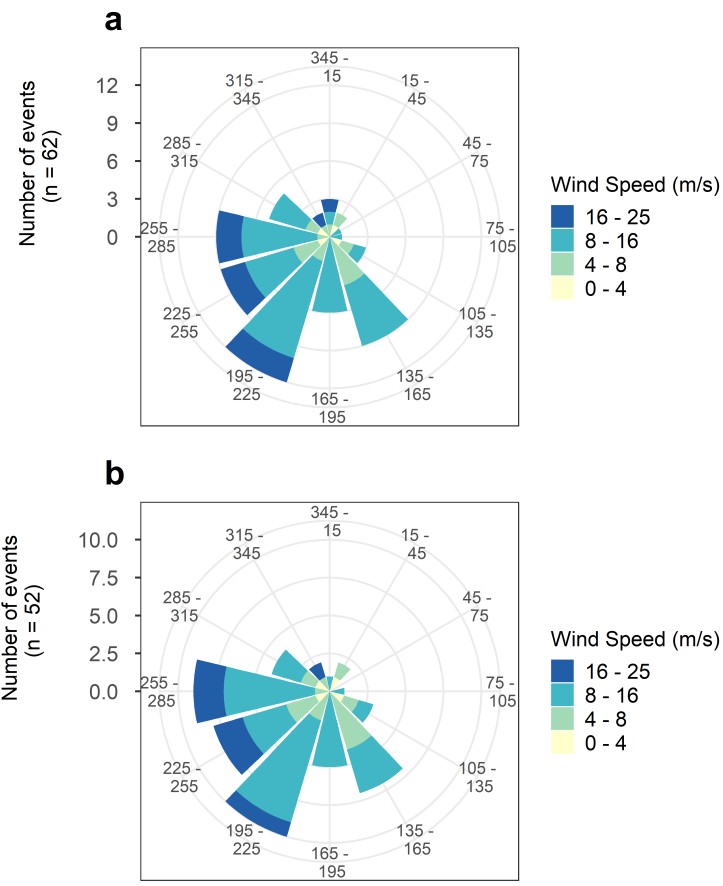

**Figure A4.** Wind direction measured by the radiosondes at 2.7 km of (**a**) the 62 snowfall events and (**b**) the 52 snowfall events coinciding with running mean $RH_{ice} \geq 97\%$ throughout the lower 2.7 km (see Sect. 3.2 for criterion).

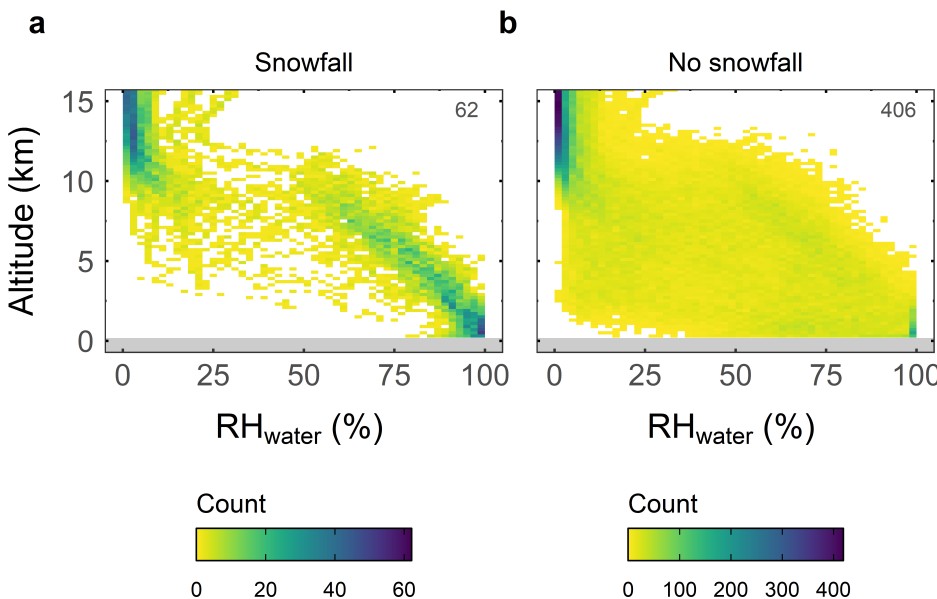

**Figure A5.** Same plot as Fig. 5, but for RH$_{water}$.

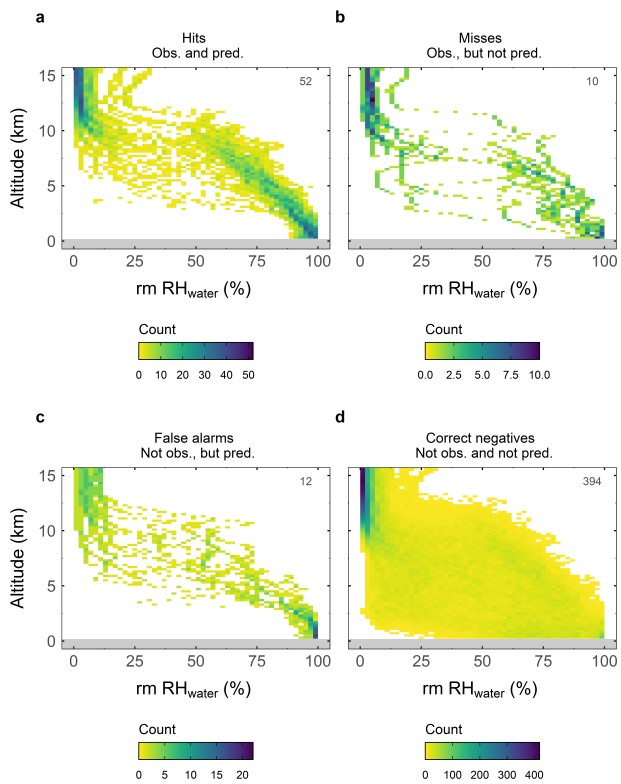

**Figure A6.** Same plot as Fig. 7, but for RH$_{water}$.

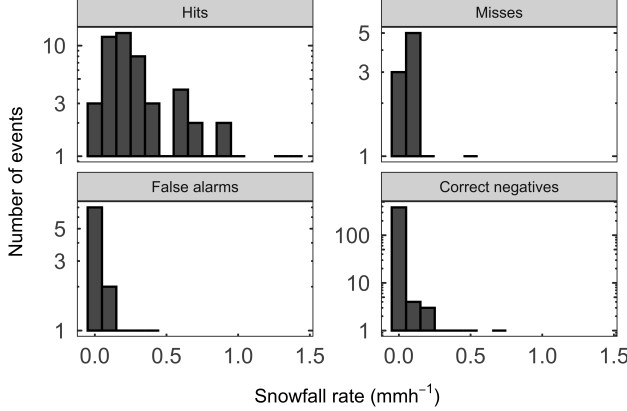

**Figure A7.** Snowfall rate by intervals measured optically during the events grouped by *hits*, *misses*, *false alarms*, and *correct negatives* based on the criterion running mean RH$_{ice}$ $\geq$ 97% throughout the lower 2.7 km to predict snowfall. The y-axis scale for each panel is different.

**Table A1.** Details of scores that compare observed versus predicted events. The scores are calculated with the help of hits (H), misses (M), false alarms (F), correct negatives (C), and the total number of events (Total).

| Abbreviation | Name | Formula | | Range | Ideal value | Reference |
|---|---|---|---|---|---|---|
| ACC | Accuracy | $\dfrac{H+C}{Total}$ | (A1) | (0, 1) | 1 | Bennett et al. (2013) |
| CSI | Critical success index | $\dfrac{H}{H+M+F}$ | (A2) | (0, 1) | 1 | Bennett et al. (2013) |
| HSS | Heidke skill score | $2\dfrac{H\cdot C - F\cdot M}{(H+M)(M+C)+(H+F)(F+C)}$ | (A3) | (0, 1) | 1 | Hyvärinen (2014) |

**Table A2.** Date and time for each of the 52 snowfall events coinciding with running mean $RH_{ice} \geq 97\%$ throughout the lower 2.7 km.

| Number | Date and time (UTC) | Number | Date and time (UTC) |
|--------|---------------------|--------|---------------------|
| 1 | 08/03/2019 23:30 | 27 | 16/12/2019 11:30 |
| 2 | 09/03/2019 11:30 | 28 | 16/12/2019 23:30 |
| 3 | 09/03/2019 23:30 | 29 | 17/12/2019 11:30 |
| 4 | 16/03/2019 23:30 | 30 | 21/12/2019 11:30 |
| 5 | 17/03/2019 11:30 | 31 | 21/12/2019 23:30 |
| 6 | 17/03/2019 23:30 | 32 | 29/12/2019 23:30 |
| 7 | 18/03/2019 11:30 | 33 | 30/12/2019 11:30 |
| 8 | 18/03/2019 23:30 | 34 | 16/01/2020 23:30 |
| 9 | 21/03/2019 23:30 | 35 | 18/01/2020 23:30 |
| 10 | 29/03/2019 11:30 | 36 | 28/01/2020 23:30 |
| 11 | 04/04/2019 11:30 | 37 | 29/01/2020 11:30 |
| 12 | 30/04/2019 23:30 | 38 | 29/01/2020 23:30 |
| 13 | 25/10/2019 23:30 | 39 | 31/01/2020 11:30 |
| 14 | 26/10/2019 11:30 | 40 | 01/02/2020 11:30 |
| 15 | 30/10/2019 23:30 | 41 | 01/02/2020 23:30 |
| 16 | 02/11/2019 11:30 | 42 | 02/02/2020 11:30 |
| 17 | 02/11/2019 23:30 | 43 | 07/02/2020 11:30 |
| 18 | 11/11/2019 23:30 | 44 | 11/02/2020 11:30 |
| 19 | 13/11/2019 23:30 | 45 | 15/02/2020 11:30 |
| 20 | 14/11/2019 11:30 | 46 | 21/02/2020 11:30 |
| 21 | 28/11/2019 11:30 | 47 | 08/03/2020 23:30 |
| 22 | 28/11/2019 23:30 | 48 | 10/03/2020 11:30 |
| 23 | 09/12/2019 11:30 | 49 | 11/03/2020 11:30 |
| 24 | 11/12/2019 11:30 | 50 | 12/03/2020 23:30 |
| 25 | 13/12/2019 11:30 | 51 | 15/03/2020 23:30 |
| 26 | 15/12/2019 23:30 | 52 | 31/03/2020 11:30 |