# Peer review of "Snowfall in Northern Finland derives mostly from ice clouds"

_Atmospheric Chemistry and Physics, 2022_

## Author Comment (AC1)

First, we would like to thank the two anonymous Reviewers for having carefully read the manuscript and for providing their helpful and constructive reviews, which improved our manuscript. Point-by-point replies to the comments are here below.

For clarity and easy visualization, the Referee's comments are shown from here on in black.

> The authors' replies are in blue font with an increased indent below each of the referee's statements. The Line numbers (L.) in our responses refer to the unrevised manuscript.

> The relevant changes in the revised manuscript are below in green.

**Authors' response to anonymous Referee #3 (https://doi.org/10.5194/acp-2022-98-RC1)**

**Overall Quality**

This manuscript utilizes a merged-instrument approach to characterize precipitating ice particle habits at a remote site in inland Finlind. Primarily using 12-hourly soundings and the Multi-Angle Snowflake camera (MASC), the study determines via knowledge of ice particle history and growth regimes that approximately three-quarters of ice particles originate from cloud layers with top temperatures outside of the mixed-phase region (i.e., sub-liquid RH saturation [<99%]), suggesting that the majority of cloud layers are fully glaciated. Using an empirical formulation, they finally determine that the number of ice nucleating particles (INP) were likely sufficient to explain heterogenous ice production, suggesting an inactive ice multiplication mechanism (outside of possible collisions). Overall, the manuscript is of excellent quality in terms of science, documentation, figures, and structure. The authors clearly made a significant effort to explain their data processing in a concise manner. After addressing a few specific comments and technical corrections, I recommend this manuscript pursue publication in ACP.

> We thank the referee for reviewing our manuscript. We appreciate the positive feedback and helpful comments.

**Specific Comments**

Fig 6. & ~Line 183: I would point out to the reader that the color-scales on each panel are different.

> Thank you for commenting on this. We have added the following sentence in the caption of Fig. 6. We also added a sentence in captions of Fig. 4 and Fig. A6, which are figure that have had similar issues.

> The color scale ranges from zero to the total number of events for each group, so the color scale for each panel is different.

Line 159 & Fig. 3: What exactly is "visibility"? If it is similar to cloud base height, then these are an order of magnitude off. It would also be good to mention how cloud base height was detected within the instrumentation at the site. If it is a nm-wavelength active remote sensor, then I would expect my interpretation of visibility to closely optically correspond with cloud
base height.

The cloud base height was measured by Vaisala CT25K ceilometer (Ceilometer
CT25K User's Guide, Vaisala, available at: https://www.rish.kyoto-
u.ac.jp/ear/ceilometer/ct25k.pdf, last access: 11 August 2022). The visibility was
measured by Vaisala FD12P with an optical forward-scatter sensor that sees fog and
precipitation particles (see
https://www.livedata.se/images/Vaisala/Nederbord/FD12P.pdf, last access: 11
August 2022). The visibility measurement range is 10 to 50 000 m. This is basically
documented in L. 116 with reference to the FMI web page
(https://litdb.fmi.fi/luo0015_data.php, last access: 11 August 2022). We replaced the
"visibility" with "horizontal visibility" throughout the manuscript. Consistently, we
adapted the sentence in L. 159.

During snowfall, the horizontal visibility was on average 2020 m, the average base
height (or vertical visibility) of the lowest cloud was 213 m, […].

Fig 3: I'm confused about the sea level pressure measurements. If the station is only 179 m
ASL, these values are way too low.

Thank you very much for this valuable comment. It brought to our attention that we
have made a mistake in calculating the ground-based meteorological parameters for
the 15 minutes intervals. We have corrected this and updated the Fig. 3. Amongst
other variables, the sea level air pressure values are higher than before and now
make sense. In addition, the related values mentioned in the text (L. 154 – 160, L.
192 – 194) and Fig. A6 were corrected.

Fig 2 & Line 134: Why 15 minutes prior to sounding release? Wouldn't 15 minutes afterward
be more representative of the cloud that is producing the precipitation?

Since radiosondes were launched from the same ground station at which we
observed falling snow crystals, it was only at ground level and at the moment of
launch that both kinds of observations coincided in space and in time. Crystals
formed at any point in the profile while it was sounded, have reached ground level
downwind the station and at a later point in time. By relating the humidity profile to
crystals observed 15 minutes prior to launch we assumed that the profile is, when
sounded, still representative of what it was up to 15 minutes earlier upwind the
station. If we would have related the sounded humidity profile to crystals observed
during the 15 minutes following sounding, we would have had to assume the
sounding to be representative of the moisture profile still upwind the station, from
where crystals would arrive in the following 15 minutes. Neither assumption is
secure, but the first seemed to us more reliable than the second. Anyway, the sky
was fully cloud covered (8 octas; see L. 162) during snow events and the choice of
assumption probably makes no big difference.

Line 213: Nice conclusion!

Thank you.
**Technical Corrections**
Line 61: "automatically" should be "automatic"
Done.
Line 63: "summery" should be "summer"
Done.
Fig A1: "lowlight" should be "highlight"
We changed the wording into "The grey areas mark…".
Line 81: suggest using "length" instead of "height"
Done.
Line 94: Should "An ice particle classified" be "An ice is particle classified"?
We changed this sentence as it was a little confusing.
Unrimed ice particles correspond to riming degrees of 0 and 1, and rimed particles to
riming degrees of 2 to 5 according to Mosimann et al. (1994).
Line 153: Should "weighed" be "weighted"?
Yes, thank you. Done.
**References**

Mosimann, L., Weingartner, E., and Waldvogel, A.: An Analysis of Accreted Drop Sizes and
Mass on Rimed Snow Crystals, J. Atmos. Sci., 51, 1548 – 1558,
https://doi.org/10.1175/1520-0469(1994)051<1548:AAOADS>2.0.CO;2, 1994.

---

## Author Comment (AC2)

First, we would like to thank the two anonymous Reviewers for having carefully read the manuscript and for providing their helpful and constructive reviews, which improved our manuscript. Point-by-point replies to the comments are here below.

For clarity and easy visualization, the Referee's comments are shown from here on in black.

> The authors' replies are in blue font with an increased indent below each of the referee's statements. The Line numbers (L.) in our responses refer to the unrevised manuscript.

> The relevant changes in the revised manuscript are below in green.

**Authors' response to anonymous Referee #2 (https://doi.org/10.5194/acp-2022-98-RC2)**

The authors use several months of radiosonde soundings and coincident, ground-based hydrometeor imagery at a high-latitude station in northern Finland to infer ice formation pathways during snow events. Relative humidity (RH) profiles (both with respect to water and ice) from radiosonde data are used to develop a simplistic snow event predictor. For snow events, the authors show how imagery-based ice particle habits change as a function of RHw. Using cloud-top temperature the authors conclude that primary ice formation was the main pathway to form snow.

The study is well written and contains many useful plots. I recommend publication after resolving a few major issues.

> Thank you very much for reviewing our manuscript, the general assessment and the constructive comments.

**Major points**

The "Results" section includes a few elements of a discussion. However, I feel the study would benefit from a broader discussion that is also placed into its own section.

> Thank you for the suggestion. We have carefully weighed benefits and drawbacks of separating "Results" from a broadened "Discussion". In the end, we decided to keep both combined for better readability and also to avoid stretching interpretation. We hope to have done justice to the valuable specific issues below in the revised "Results and discussion" section.

Following points should be relevant to the reader:

- The authors start their study by mentioning the Arctic surface budget. Do the authors think the site in Finland is representative of the Artic? Or could the continental character and the influence from boreal forests (e.g., Schneider et al., 2021) mislead?

Indeed, the opening sentence may raise expectations that cannot be fully met. Therefore, we changed some sentences of our manuscript in the Abstract (L. 1, L. 2, L. 9) as well as the Introduction (L. 12 – 14, L. 20). Nevertheless, we still find the more nuance relation to Arctic studies appropriate in L. 40 onward.

- Would other (frequently used) INP parameterization lead to the same conclusions?

  To our knowledge, no aerosol properties were measured continuously at the site over the period of our study, except aerosol optical depth. However, most commonly used INP parameterizations are based on aerosol particle properties such as concentration or size distribution. Lacking such data, we are not able to use such INP parameterizations and thus do not know whether the use of other INP parameterizations would lead to the same conclusion. We added the following sentence in L. 242.

  Since the necessary aerosol properties were not measured at the site at the time period of interest, it was not possible to use other existing INP parameterisations (e.g. DeMott et al., 2015, Ullrich et al. 2017) to qualitatively evaluate the associated uncertainty. However, the here predicted INP concentrations are similar to…

- Could the high-RHw group (Fig. 8) be useful as a proxy of snow events in a warmer climate?

  Conceptually yes. Therefore, we added the following sentence to the conclusion.

  This could lead to snowfall events with a greater proportion of larger snowflakes, rimed ice particles, and such crystal habits that grow above water saturation compared to today. Rime-splintering and other secondary ice formation processes requiring liquid droplets could become more frequent, which would likely increase the ice multiplication factor.

- Is a 15 min window appropriate? How long would it take for a particle to fall from ~2.7 km?

  A compactly growing ice particle falls about 800 m during the first 30 minutes of its growth (Fukuta and Takahashi, 1999). Assuming thereafter a fall velocity of 1 m s$^{-1}$, the time it will take to reach ground from an initial height of 2.7 km is around 1 hour. Ideally, we would have used slowly descending drop sondes, dropped so far upwind that they would have arrived at the ground station together with the snow crystals they had accompanied during their growth. However, radiosondes launched from ground level travel vertically in the opposite direction of falling crystals. Therefore, temporal and spatial lags between the trajectories of radiosonde and observed crystals are unavoidable. Minimising these lags can only be achieved by choosing a short interval for crystal observation. Still, the interval has to be long enough to detect low precipitation rates. We settled for 15 minutes, which was enough to detect precipitation rates ≥ 0.01 mm h$^{-1}$ (see L. 140 – 144).

Please review the order of the figures. Figure 7 is mentioned earlier (l. 89) than Figure 2 (l.
150). The same review should be applied for supplementary figures.
Thank you. We separated Fig. 7a from Fig. 7b and moved Fig. 7a. Furthermore, Fig.
A7 and Fig. A3 are now arranged in a way that it follows the narrative. Consequently,
most Figure numbers have changed in the revised manuscript.
**Minor points**
l. 1 This sentence sticks out. Either specify "properties" and their "role" or write it more
general as "clouds" (instead of "cloud properties").
We generalized.
Clouds and precipitation play a critical role in the Earth's water cycle and energy
budget.
ll. 198-199 Perhaps show examples of unclassifiable particles.
We added some examples of invalid and undefinable particles to Fig. A7 and refer to
it in the text. Also, we show some examples of broken-off branches of dendrites (see
Figure here below).

[Figure]

**Figure A2.** Similar to Fig. 2, but with further examples of images captured by the
MASC. Aggregates of specifiable ice particle shapes are arranged in the top row.
Invalid and undefinable particle shapes as well as broken-off branches of dendrites
are shown in the bottom row.
l. 208 This sentence is redundant as the information was provided in l. 206.

Thank you for spotting the redundancy. We have changed the sentence in L. 206.

I. 225 How is cloud-top temperature obtained?

Since this question and the following one are related, we will answer them together
below.

II. 226-228 This description needs improvement and perhaps an illustration of the concept.
What is meant by "gaps" and how do you determine them?

Thank you very much for these questions. First, we have marked the 100 m thick
atmospheric layers of the radiosonde profiles with increasing height where the
running mean of $RH_{ice}$ is ≥ 100%. Such atmospheric layers are regions where ice
crystals are not sublimating and defined here as "clouds". For each cloud, we
determined the cloud bottom height, the cloud top height and cloud top temperature.
The cloud top temperature is the minimum temperature between cloud bottom and
cloud top height measured by the temperature sensor of the radiosonde. If two
clouds were on top of each other, we determined the distance between the cloud
bottom height of the upper-level cloud and the cloud top height of the lower-level
cloud. In case this distance was below a certain threshold (0.2, 0.5 or 1 km), we
considered these two clouds as being potential seeder-feeder clouds. If there were
more than two clouds on top of each other, we determined the highest seeder cloud
for which the distance threshold with increasing height holds. We assume that ice
crystals from the upper-level cloud (seeder) could potentially fall through the
unsaturated layer without completely sublimating and therefore "seeding" the lower-
level cloud (feeder). Finally, we determined the cloud top temperature of the highest
possible seeder cloud for each case. Note that we do not distinguish between cases
with a single cloud and those with feeder-seeder clouds.

We added this information to the manuscript and show an example of a case which
has had three clouds (see Figure 2 here below).

[Figure]

**Fig. 2. (a)** Running mean of five consecutive 100 m averaged $RH_{ice}$ with increasing height up to 15 km on the 9th of March 2019 at 23:30. Vertical line indicates 100% running mean $RH_{ice}$. The dots colored in blue show the values ≥ 100%. **(b)** Similar to a) except that the temperature (°C) is plotted on the x-axis. Dashed green line shows the cloud top temperature of the highest possible seeder cloud, if we consider a threshold of 0.2 and 0.5 km between potential seeder-feeder clouds. The dotted red line shows the cloud top temperature of the highest possible seeder cloud, if we consider a threshold of 1.0 km between potential seeder-feeder clouds.

While working on the revisions and re-running some code, we noticed that several cases with multiple clouds on top of each other were not handled correctly. We corrected this and updated Fig. 10, which also resulted in changes in the text. The legend of Fig. 10 has been adjusted. Note that in the revised version we don't use the word "gaps" anymore, but replaced it with "distance between clouds". Thank you for this very useful comment.

Here we define successive values of the running mean from five consecutive 100 m averaged $RH_{ice}$ with increasing height ≥ 100% as a cloud. For each cloud, we determined the cloud base height, cloud top height, and cloud top temperature. The cloud top temperature is the minimum temperature between the cloud base and cloud top measured by the radiosonde temperature sensor. When two clouds were on top of each other, we further determined the distance between the cloud top height of the lower-level cloud and the cloud base height of the upper-level cloud. If this distance was below a certain threshold (0.2, 0.5, or 1 km), we considered the two clouds as potential seeder-feeder clouds. If there were more than two clouds on top of each other, we determined the highest seeder cloud for which the distance threshold with increasing height holds. Finally, we determined the cloud top temperature of the highest seeder cloud for each case (also described as cloud top temperature from here on). Hence, we take into account that up to a certain (vertical) distance between clouds, ice crystals from the upper cloud (seeder) could fall through the unsaturated layer without fully sublimating, thus seeding the lower cloud (feeder). It is noteworthy that our threshold for seeder-feeder consideration is based only on the distance between clouds. However, whether an ice crystal fully sublimates between two clouds depends on several factors such as the crystal's size and habit when entering unsaturated conditions or by how much conditions are unsaturated. In the absence of in-cloud crystal information, we cannot make detailed calculations. To compensate for this uncertainty, we show cloud top temperature estimates for three different distance thresholds. Note that we do not distinguish between cases with a single cloud and those with seeder-feeder clouds.

[Figure]

**Figure 11.** (**a**) The cloud top temperatures of the highest possible seeder-feeder cloud in 4 °C intervals of the 52 events coinciding with running mean $RH_{ice} \geq 97\%$ throughout the lower 2.7 km. The cloud top temperature was derived using the radiosonde measurements which is described in detail in Sect. 3.5. We used the following thresholds for the distance between clouds to be considered as seeder-feeder clouds: ≤ 0.2 km (orange), ≤ 0.5 km (pink), and ≤ 1.0 km (purple). The fraction of events with cloud top temperatures above −38 °C is given in percent next to the dashed line. This is an estimation of the fraction of events for which the first ice crystals were likely formed via heterogeneous freezing. (**b**) Density of the INP concentration for the fraction of events with cloud top temperatures above −38 °C, using the different thresholds to account for seeder-feeder clouds cloud top height criteria (in color) as shown in panel a. The respective median concentrations are shown by the dashed lines.

ll. 244-247 This seems highly relevant and should be shown as its own plot.

Thank you. We agree with the reviewer and made a new plot, which is shown here below. We discuss the related results in more detail in the last paragraph of Section 3.5 of the revised manuscript.

[revised manuscript text omitted]

---

## Author Response (AR2)

We would like to thank the Editor and the anonymous Reviewer for having carefully read the revised version of the manuscript. We would like to thank the Editor for providing helpful comments, which improved the manuscript. Point-by-point replies to the comments are here below.

For clarity and easy visualization, the Editor's comments are shown from here on in black.

> The authors' replies are in blue font with an increased indent below each of the referee's statements. The Line numbers (L.) in our responses refer to the unrevised manuscript.

> The relevant changes in the revised manuscript are below in green.

**Comments to the author:**
I would like to thank the authors for incorporating the suggestions made by the reviewers. The revised version looks pretty good and it is almost ready for publication; however, I have the following additional and final comments before I can accept the manuscript.

**Minor/Technical Comments:**

Lines 8 and 274: Should it be "thirds"?

> Yes.

Line 17:…-38°C and RHi >145%...

> Thank you. We adapted this.

> …−38°C when the relative humidity with respect to ice is above 145%...

Line 18: …temperatures and lower RHi vales…

> Thanks.

Line 19: "water vapor deposition"

> Done.

Lines 34-35: …with respect to ice (RHice) and liquid water (RHwater)…

> Done.

Line 65 and Figure 2: "Of note" sounds a bit strange to me.

> Deleted.

Lines 65-68: Do the authors know how these changes could have impact the delivered results compared if the "original" version was used?

> These slight modifications, which are unrelated to the main instrument components, should not have impacted the results. We added a sentence.

Since the main components of the instrument, including the cameras, have been kept in their original condition, we expect that our instrument provides the same results as an unmodified MASC.

Line 68: What do the authors mean with "on the instrument field"?

We meant measurement field and reformulated two sentences.

The instrument was installed at two meters above ground next to other field instruments that are part of…

L. 121: In the WMO SPICE measurement field,…

Line 89: SUCH with an undefinable habit, SUCH as broken…

Thank you.

Line 91:…columns, and rosettes)

Done.

Lines 109-110:…profiles of RHice were…

Done.

Lines 114-115:…temperature, RHice,…

Done.

Line 137: "Radiosondes were…..UTC". This was already mentioned above. I suggest to delete it.

O.K.

Line 151: What do the authors mean with "eventual riming"?

Thank you. We meant riming degree.

Line 154: I suggest to replace "meteorological parameters on ground" by "Ground-based meteorological parameters"

Done.

Line 166:…i.e. RHwater, pressure…

Done.

Line 169: The RHice of the profiles…

Done.

Line 170: "meters"

Thanks.

Lines 172-173: RHwater

Done.

Line 243: how about "by the degree of unsaturated air"

Sounds good. Thank you.

Lines 250-252: I find this a bit speculative as the composition of the INPs was not monitored in the present study. I suggest to change/soften the tone of this conclusion.

True. We reformulated it and say "we speculate that…"

Line 251: INPs defined above.

O.K.

Line 264: "condensation freezing mode"

Thanks.

Line 283: Should it be "in relation to the temperatures observed"?

Yes.

Figure 1: In the same line you have "LOCATION" and "LOCATED". Please fix this.

Done.

Figure 8: What do the authors mean with "as shown in a"?

Thanks. We now refer to the appropriate Figure.

Figure 12: Should it be ""snowfall events"?

Yes.

Figure A1. Delete "the" before "pressure".

Done.

Table A2: Date and time FOR each…

Thank you.